# CLIENT2VEC: IMPROVING FEDERATED LEARNING BY DISTRIBUTION SHIFTS AWARE CLIENT INDEXING

## ABSTRACT

Federated Learning (FL) is a privacy-preserving distributed machine learning paradigm. Nonetheless, the substantial distribution shifts among clients pose a considerable challenge to the performance of current FL algorithms. To mitigate this challenge, various methods have been proposed to enhance the FL training process. This paper endeavors to tackle the issue of data heterogeneity from another perspective—by improving FL algorithms prior to the actual training stage. Specifically, we introduce the Client2Vec mechanism, which generates a unique client index for each client before the commencement of FL training. Subsequently, we leverage the generated client index to enhance the subsequent FL training process. To demonstrate the effectiveness of the proposed Client2Vec method, we conduct three case studies that assess the impact of the client index on the FL training process. These case studies encompass enhanced client sampling, model aggregation, and local training. Extensive experiments conducted on diverse datasets and model architectures show the efficacy of Client2Vec across all three case studies. Our code will be publicly available.

## 1 INTRODUCTION

Federated Learning (FL) is an emerging machine learning paradigm that preserves clients' privacy by only exchanging model parameters between clients and server, and maintains the local data not exchanged. As the de facto algorithm in FL, FedAvg (McMahan et al., 2016) proposes to use local SGD to improve the communication efficiency of FL. However, the non-i.i.d. nature of local distributions significantly reduces the performance of FL algorithms (Lin et al., 2020b; Karimireddy et al., 2020b; Li et al., 2020). Despite the great success of existing methods in addressing the non-i.i.d. problem in FL (Li et al., 2021; Acar et al., 2020), most existing studies center on the training process of FL by improving the key stages of the FL training, such as client sampling (Fraboni et al., 2021; Luo et al., 2022; Wang et al., 2023), model aggregation (Wang et al., 2019; Lin et al., 2020a; Chen et al., 2023), and local training (Li et al., 2020; 2021).

An additional line of research in FL aims to find efficient methods to improve the performance before the training stage. Yet only a limited number of works exist, either utilizing dataset distillation before the FL training (Yang et al., 2023), or generating global shared synthetic pseudo-data (Guo et al., 2023b; Tang et al., 2022). Despite promising, these approaches incur additional computation costs on local devices with a limited number of applicable scenarios and are incompatible with other FL training stages like client sampling and model aggregation.

Taking inspiration from the Word2Vec technique in Natural Language Processing (NLP) tasks (Mikolov et al., 2013) and domain indexing in Domain Generalization tasks (Xu et al., 2022), we introduce a novel mechanism below, namely Client2Vec. Client2Vec generates an index vector for each client, serving as their identity by incorporating information about the client's local data distribution. These vectors are used to measure label and feature distribution shifts among clients, seamlessly integrate into the FL training pipeline, and allows efficiently (1) operating independently of the FL training process, (2) combining with existing FL methods, while imposing a minimal additional computational load on local devices, and (3) enhancing FL training performance throughout all stages.

Our contributions can be summarized as follows:

- We explore a novel mechanism in FL called Client2Vec, which involves creating an index vector for each client before the FL training phase. These vectors incorporate information about the local distribution of clients and subsequently enhance the FL training process.
- We present the Distribution Shifts Aware Index Generation Network (DSA-IGN), a network specifically designed to generate the client index prior to FL training. Our visualization results demonstrate the effectiveness of the client index in measuring the similarities in clients' local distributions.
- We conduct three case studies, including client sampling, model aggregation, and local training, to illustrate the potential and effectiveness of the generated client index. Our experiments, conducted on various datasets and model architectures, consistently demonstrate significant performance improvements with the use of Client2Vec.

## 2 RELATED WORKS

**Information sharing in FL.** Federated Learning (FL) is a distributed training methodology where local data is retained and not exchanged between the central server and clients (Li et al., 2020; Karimireddy et al., 2020b;a; Guo et al., 2023b; Jiang & Lin, 2023). FedAvg (McMahan et al., 2016; Lin et al., 2020b), a foundational algorithm, uses local Stochastic Gradient Descent (local SGD) to reduce communication. Nevertheless, the performance of FL algorithms is substantially impeded by distribution shifts among clients. To address distribution shift in FL, existing works share local distribution statistics (Shin et al., 2020; Zhou & Konukoglu, 2022), data representations (Hao et al., 2021; Tan et al., 2022), and prediction logits (Chang et al., 2019; Luo et al., 2021). FedMix (Yoon et al., 2020) and FedBR (Guo et al., 2023b) enhance local training with privacy-protected augmentation data. VHL (Tang et al., 2022) employs randomly initialized generative models to produce virtual data, regularizing local features to closely align with those of same-class virtual data. FedFed (Yang et al., 2023) proposes a dataset distillation method, amalgamating distilled datasets into all clients' local datasets to mitigate distribution shifts. Compared to existing methods, Client2Vec has the following advantages: (1) decouples index generation from FL training, reducing FL training load; (2) generates one client index per client, improving efficiency; (3) contributes to the entire FL training process, including client sampling, model aggregation, and local training.

**Domain indexing.** Domain Generalization (DG) tackles cross-domain generalization by generating domain-invariant features. While conventional DG methods strive to make a data point's latent representation independent of its domain identity using a one-hot vector (Ganin et al., 2016; Tzeng et al., 2017; Zhao et al., 2017), recent studies propose using real-value scalars or vectors as domain indices to improve performance (Wang et al., 2020; Xu et al., 2021). However, obtaining domain indices may be impractical. To address this, Xu et al. (2022) introduced variational domain indexing (VDI) to infer domain indices without prior knowledge. Yet, challenges arise when applying VDI to FL due to communication costs, privacy concerns, and neglect of label shifts. Further discussions on related works can be found in Appendix B.

## 3 CLIENT2VEC: DISTRIBUTION SHIFTS AWARE CLIENT INDEXING

In this section, we introduce Client2Vec, a mechanism that generates an index vector for each client, representing their identity and incorporating information about their local distribution. The client index, which considers both label and feature distribution shifts, is defined in Section 3.1. We present the Distribution Shifts Aware Index Generation Network (DSA-IGN) in Section 3.2, which generates the client index based on the specified criteria. Visualization examples of the generated client index are provided in Section 3.3.

### 3.1 CLIENT INDEX

We consider the FL setting with $M$ clients, where each client $i$ possesses $N_i$ data samples. The $j$-th data sample of client $i$ is represented as $(\mathbf{x}_{i,j}, y_{i,j})$.

**Sample index $\mathbf{u}$ and Client index $\beta$.** To ensure that the client index conveys information about all data samples within the client, we first generate the sample index $\mathbf{u}_{i,j}$ as the index vector for the data

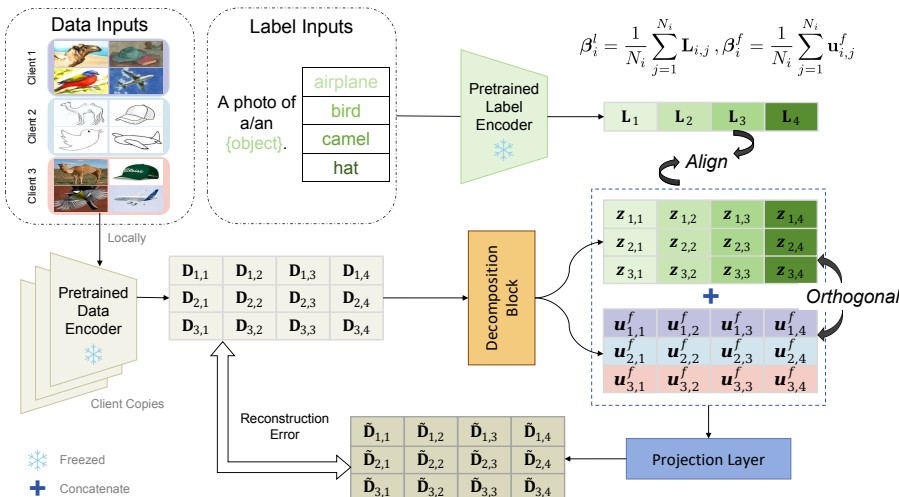

Figure 1: **Illustration of the DSA-IGN Workflow:** The local data from clients, denoted as $(\mathbf{x}_{i,j}, y_{i,j})$, undergo encoding by the CLIP encoders, resulting in the transformation to $(\mathbf{D}_{i,j}, \mathbf{L}_{i,j})$ before the index generation process. The CLIP image embedding $\mathbf{D}_{i,j}$ is then split into a data encoding $\mathbf{z}_{i,j}$ and a sample feature index $\mathbf{u}_{i,j}^f$. The $\mathbf{z}_{i,j}$ and $\mathbf{u}_{i,j}^f$ are then concatenated and projected to $\tilde{\mathbf{D}}_{i,j}$ to reconstruct $\mathbf{D}_{i,j}$. Lastly, client label index $\boldsymbol{\beta}_i^l$ and client feature index $\boldsymbol{\beta}_i^f$ are obtained by averaging $\mathbf{L}_{i,j}$ and $\mathbf{u}_{i,j}$, respectively.

sample $(\mathbf{x}_{i,j}, y_{i,j})$. The client index $\boldsymbol{\beta}_i$ is then computed as the average of all data samples for client $i$: $\boldsymbol{\beta}_i = \frac{1}{N_i} \sum_{j=1}^{N_i} \mathbf{u}_{i,j}$.

For FL scenarios where feature and label shifts occur simultaneously, the sample index $\mathbf{u}_{i,j}$ consists of two parts: sample feature index $\mathbf{u}_{i,j}^f \in \mathbb{R}^{d_i}$ and sample label index $\mathbf{u}_{i,j}^l \in \mathbb{R}^{d_i}$, encoding the feature and label information of the data sample $(\mathbf{x}_{i,j}, y_{i,j})$, respectively. Similarly, the client index $\boldsymbol{\beta}_i$ is represented as $\boldsymbol{\beta}_i = [\boldsymbol{\beta}_i^f; \boldsymbol{\beta}_i^l] \in \mathbb{R}^{2d_i}$, where $\boldsymbol{\beta}_i^f \in \mathbb{R}^{d_i}$ is the client feature index, and $\boldsymbol{\beta}_i^l \in \mathbb{R}^{d_i}$ is the client label index.

However, obtaining client and sample indices may not always be trivial in practice. To address this, we extend the domain index idea in Xu et al. (2022) and define the expected properties of sample index $\mathbf{u}_{i,j}$ and client index $\boldsymbol{\beta}_i$ below.

**Definition 3.1** (Sample Index). *Given the data sample $(\mathbf{x}_{i,j}, y_{i,j})$ and its corresponding encoding $\mathbf{z}_{i,j}$, which encode information that can be used to predict $y_{i,j}$, the sample index $\mathbf{u}_{i,j}$ of data sample $(\mathbf{x}_{i,j}, y_{i,j})$ is expected to satisfy the following properties:*

- ***Independence between $\mathbf{u}_{i,j}^f$ and $\mathbf{z}_{i,j}$.*** *Sample feature index $\mathbf{u}_{i,j}^f$ is independent of client-invariant data encoding $\mathbf{z}_{i,j}$. This aims to encourage the sample feature index $\mathbf{u}_{i,j}^f$ to encode the client-dependent information—specifically, the distinct information about the client's local distribution, and unrelated to label prediction.*
- ***Maximizing information in $\mathbf{u}_{i,j}^l$ and $\mathbf{z}_{i,j}$ for label prediction.*** *The data encoding $\mathbf{z}_{i,j}$ and sample label index $\mathbf{u}_{i,j}^l$ should contain as much information as possible to predict label $y$.*
- ***Information Preservation of $\mathbf{u}_{i,j}^f$ and $\mathbf{z}_{i,j}$.*** *Data encoding $\mathbf{z}_{i,j}$ and sample feature index $\mathbf{u}_{i,j}^f$ preserves as much information as possible to recover data $\mathbf{x}_{i,j}$.*

**Remark 3.2.** *Different from the definitions of domain index outlined in Xu et al. (2022) (refer to Definition C.1), Definition 3.1 encompasses both label and feature distribution shifts by introducing the sample label index $\mathbf{u}_{i,j}^l$. Additionally, generating domain-level index in Definition C.1 requires to gather all sample indices $\mathbf{u}_{i,j}$ from various data sources. We simplify this procedure and calculate the client index as $\boldsymbol{\beta}_i = \frac{1}{N_i} \sum_{j=1}^{N_i} \mathbf{u}_{i,j}$.*

## 3.2 GENERATING CLIENT INDEX

In this section, we describe the generation of $\boldsymbol{\beta}_i$ and $\mathbf{u}_{i,j}$ based on Definition 3.1, as depicted in Figure 1. We use image datasets for clarity, and details for language datasets can be found in Appendix D.1.

**Encoding data using CLIP.** Since the client index, denoted as $\boldsymbol{\beta}_i$, is computed as an average across sample indices $\mathbf{u}_{i,j}$, the primary challenge in generating the client index lies in generating these sample indices $\mathbf{u}_{i,j}$. According to Definition 3.1, we can observe that the sample label index $\mathbf{u}_{i,j}^l$ solely encodes label information, while the sample feature index encodes image feature information independently of label information. Consequently, to generate the sample index, we must devise a method that maps label information and image information into the same space, facilitating the extraction of label-dependent sample label index $\mathbf{u}_{i,j}^l$ and label-independent sample feature index $\mathbf{u}_{i,j}^f$.

We propose to leverage CLIP (Radford et al., 2021) to ease the index generation process. CLIP is a pre-trained cross-modality model that contains an image encoder and a text encoder, aligning image and text embedding. As shown in Figure 1, for a given input image-label pairs $(\mathbf{x}_{i,j}, y_{i,j})$, we utilize a CLIP image encoder to produce image embedding $\mathbf{D}_{i,j}$ and a text encoder to generate label embedding $\mathbf{L}_{i,j}$:

- *Label Embedding* $\mathbf{L}_{i,j}$: The $\mathbf{L}_{i,j}$ only encodes label descriptions such as "A photo of a/an {object}". Therefore, we use $\mathbf{L}_{i,j}$ as the embedding that only contains label information, which is naturally invariant among clients.
- *Image embedding* $\mathbf{D}_{i,j}$: The $\mathbf{D}_{i,j}$ is extracted from the whole image, serving as a compact feature containing both client-independent (label information) and client-specific (background, style, etc.) features.

Consequently, the original local dataset $\mathcal{D}_i = \{(\mathbf{x}_{i,j}, y_{i,j})\}$ is transformed into the CLIP embedding set $\mathcal{E}_i = \{(\mathbf{D}_{i,j}, \mathbf{L}_{i,j})\}$. The $\mathcal{E}_i$ is then utilized to generate the index. Note that other cross-modality models, such as BLIP (Li et al., 2022) and BLIP2 (Li et al., 2023a), can also align vision and language like CLIP. Exploring the effectiveness of these models could be a valuable future research direction.

**Generating sample indices using CLIP embedding and DSA-IGN.** Given that $\mathbf{L}_{i,j}$ only encodes label information, we directly set $\mathbf{u}_{i,j}^l = \mathbf{L}_{i,j}$. On the contrary, by Definition 3.1, the sample feature index $\mathbf{u}_{i,j}^f$ needs to encode label-invariant client-specific information. Thus, we decompose $\mathbf{D}_{i,j}$—which contains both label and client-specific information—to generate $\mathbf{u}_{i,j}^f$ while isolating the client-specific information from $\mathbf{D}_{i,j}$.

The entire process of generating the sample feature index is achieved by training the Distribution Shifts Aware Generation Network (DSA-IGN). The training process of DSA-IGN includes three components, corresponding to the three rules in Definition 3.1:

- *Decompose CLIP image embedding* $\mathbf{D}_{i,j}$. We decompose $\mathbf{D}_{i,j}$ to sample feature index $\mathbf{u}_{i,j}^f$ and data encoding $\mathbf{z}_{i,j}$. Regularization is applied to ensure orthogonality between $\mathbf{u}_{i,j}^f$ and $\mathbf{z}_{i,j}$, ensuring their independence. The decomposition block can be any non-linear neural network architecture, and we use a three-layer transformer encoder as the decomposition block for the sake of simplicity[1].
- *Aligning the data encoding* $\mathbf{z}_{i,j}$ *and* $\mathbf{L}_{i,j}$ *to ensure label sensitivity of* $\mathbf{z}_{i,j}$. We force $\mathbf{z}_{i,j}$ to have a large cosine similarity with the label embedding $\mathbf{L}_{i,j}$, ensuring that $\mathbf{z}_{i,j}$ encodes as much information as possible to predict the label $y_{i,j}$.
- *Reconstruct CLIP image embedding* $\mathbf{D}_{i,j}$. To ensure that $\mathbf{u}_{i,j}^f$ and $\mathbf{z}_{i,j}$ retain all the information from the CLIP image embedding $\mathbf{D}_{i,j}$, we utilize $\mathbf{u}_{i,j}^f$ and $\mathbf{z}_{i,j}$ for the purpose of reconstructing $\mathbf{D}_{i,j}$. In detail, we begin by concatenating $\mathbf{u}_{i,j}^f$ and $\mathbf{z}_{i,j}$, followed by projecting the resultant vector onto $\tilde{\mathbf{D}}{i,j}$, and subsequently minimizing the distance between the reconstructed embedding

---

[1] Each transformer encoder layer will have 8 attention heads and the dimension of the model is 32. More details about the network architectures of DSA-IGN can be found in Appendix D.2.

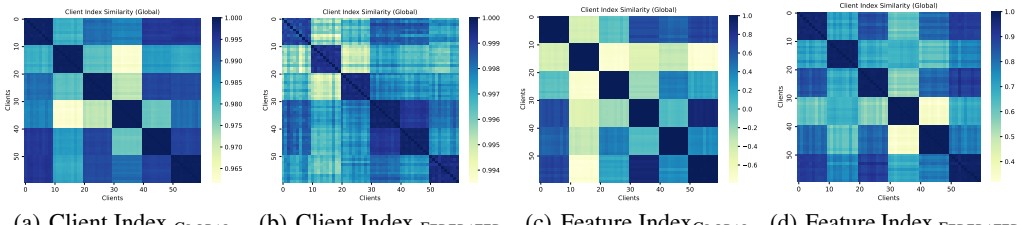

(a) Client Index $_{\text{GLOBAL}}$  (b) Client Index $_{\text{FEDERATED}}$  (c) Feature Index$_{\text{GLOBAL}}$  (d) Feature Index $_{\text{FEDERATED}}$

Figure 2: **Visualization of index similarities between clients.** We illustrate the similarities of client index $\boldsymbol{\beta}_i$ and client feature index $\boldsymbol{\beta}_i^f$ between clients. Results including both GLOBAL and FEDERATED training strategies are reported. Ideally, clients in the same domain should share a similar client index, resulting in dark diagonal blocks.

$\tilde{\mathbf{D}}i, j$ and the original CLIP image embedding $\mathbf{D}_{i,j}$. Experiments on various projection layer architectures can be found in Appendix D.3.

**Objective functions of DSA-IGN.** We define the following objective function:

$$\mathcal{L}(\mathbf{u}_{i,j}^f, \mathbf{z}_{i,j}, \mathbf{D}_{i,j}, \mathbf{L}_{i,j}) = \underbrace{\mathcal{L}_{\text{div}}(\mathbf{u}_{i,j}^f)}_{\text{Stable Training}} + \underbrace{\mathcal{L}_{\text{sim}}(\mathbf{z}_{i,j}, \mathbf{L}_{i,j}) + \mathcal{L}_{\text{orth}}(\mathbf{u}_{i,j}^f, \mathbf{z}_{i,j}) + \mathcal{L}_{\text{recon}}(\mathbf{u}_{i,j}^f, \mathbf{z}_{i,j}, \mathbf{D}_{i,j})}_{\text{Following three components}},$$

where we use $\mathcal{L}_{\text{sim}}$, $\mathcal{L}_{\text{orth}}$, and $\mathcal{L}_{\text{recon}}$, corresponding to three components, to generate sample feature indexes as defined in Definition 3.1. Additionally, we introduce $\mathcal{L}_{\text{div}}$ to improve training stability. In detail,

- $\mathcal{L}_{sim}$ *ensures label sensitivity for* $\mathbf{z}_{i,j}$. It is defined as $\mathcal{L}_{sim}(\mathbf{z}_{i,j}, \mathbf{L}_{i,j}) = 1 -$ cosine similarity$(\mathbf{z}_{i,j}, \mathbf{L}_{i,j})$, promoting a high cosine similarity between $\mathbf{z}_{i,j}$ and $\mathbf{L}_{i,j}$.
- $\mathcal{L}_{orth}$ *guarantees independence between* $\mathbf{u}_{i,j}^f$ *and* $\mathbf{z}_{i,j}$. It is defined as $\mathcal{L}_{\text{orth}} = \lVert \mathbf{Z}\mathbf{U}^T \rVert 1$, where $\mathbf{Z} = [\mathbf{z}_{i,j}]^T$ and $\mathbf{U} = [\mathbf{u}_{i,j}^f]^T$.
- $\mathcal{L}_{recon}$ *for information preservation.* It is defined as the mean squared distance between the reconstructed embedding $\tilde{\mathbf{D}}_{i,j}$ and the original CLIP image embedding $\mathbf{D}_{i,j}$.
- $\mathcal{L}_{\text{div}}$ is introduced to ensure stable training outcomes by promoting diversity in $\mathbf{u}_{ij}^f$ across different samples within the same batch. This is important because insufficient training epochs can result in identical $\mathbf{u}_{ij}^f$ values across all data samples. $\mathcal{L}_{\text{div}} = \frac{1}{B} \sum_{j=1}^B \log(\sum_{k \neq j} \exp(\text{cos-sim}(\mathbf{u}_{i,j}^f, \mathbf{u}_{i,k}^f)))$ is designed to promise diversity between $\mathbf{u}_{i,j}^f$, and it is similar in concept to SimCLR (Chen et al., 2020), focusing on negative pairs. $B$ is the batch-size here.

**Optimizing Client2Vec.** We consider two training strategies, namely GLOBAL and FEDERATED explained below:

- GLOBAL: Each client uploads one batch (128 samples) of $(\mathbf{D}_{i,j}, \mathbf{L}_{i,j})$ pairs to the server. The server trains the DSA-IGN using the collected pairs from all clients and then sends the trained DSA-IGN to clients for generating client index.
- FEDERATED: The DSA-IGN is trained using all clients' local data through FedAvg. In each communication round, the server randomly selects 10% of clients, and the local epoch number is set to 10.

### 3.3 VISUALIZATION EXAMPLES OF CLIENT INDEX

In this section, we visualize some examples of the generated client index on the DomainNet dataset (Peng et al., 2019a). DomainNet contains data from 6 different domains, where data belonging to different domains have significant feature shifts. We chose 50 out of 345 available classes, with each domain randomly divided into 10 clients. Then the 6 domains will result in a total of 60 clients. Clients 0 to 9 correspond to the first feature domain, clients 10 to 19 to the second, and so forth, with clients 51 to 59 representing the last feature domain. Clients from various domains experience feature shifts and possess varying sample sizes due to differences in the number of samples within each domain.

**Similarity of client indices among different clients.** As the distribution shifts among clients mainly come from the feature shifts among domains, in Figure 2 we depict the similarities of the client index $\boldsymbol{\beta}_i$ and the client feature index $\boldsymbol{\beta}_i^f$ across clients, employing both GLOBAL and FEDERATED training strategies. We have the following observations:

- *Clients sharing the same feature domain show similar client indices.* We observe the similarities between client index $\boldsymbol{\beta}_i$ and client feature index $\boldsymbol{\beta}_i^f$ approach 1.0 for clients within the same feature domain. Conversely, clients belonging to different feature domains have large distances regarding the client feature index $\boldsymbol{\beta}_i^f$. This highlights the effectiveness of our method in learning meaningful information about clients' local distribution.
- *Both the GLOBAL and FEDERATED training strategies produce client indices that encode meaningful information about client distribution.* We note that both the GLOBAL and FEDERATED training strategies can generate similar client indices for clients with the same feature domain. Moreover, the indices generated by the GLOBAL strategy exhibit a more clear boundary among domains.

## 4 IMPROVING FL VIA CLIENT2VEC

In this section, we showcase improving the FL training process by leveraging the trained client index $\boldsymbol{\beta}_i$. Specifically, we explore three case studies aimed at refining crucial aspects of the FL training pipeline: client sampling, model aggregation, and local training.

The intuition of using client index comes from the following reasons:

- **The client index can measure the distance between clients, which helps for improving the client sampling (case study 1) and model aggregation (case study 2) stages.** As depicted in Figure 3, clients with similar distributions tend to exhibit smaller distances between their respective Client Indexes. The rationale for utilizing this distance information is derived from the analysis presented in CyCP (Cho et al., 2023) and empirical findings in class incremental learning (He et al., 2022). These sources indicate that reduced distances among client groups sampled in consecutive rounds contribute to improved performance.
- **The client feature index is orthogonal to the ideal client-invariant features, which are defined as distribution bias in the domain generalization area. This orthogonal property contributes to improved local training (case study 3).** According to Definition 3.1, the Client Index is independent of client-invariant features, which are highly desirable in practical applications. Therefore, the objective of enhancing local training is to ensure that the model features maintain their independence from the Client Indexes.

We will introduce the details of the three case studies in the following parts of this section.

**Case study 1: improved client sampling.** The client index, $\boldsymbol{\beta}_i$, is a natural metric for measuring distance between clients. Motivated by the theoretical findings in CyCP (Cho et al., 2023) and empirical observations in class incremental learning (He et al., 2022)—where a smaller distance among client groups sampled in adjacent rounds improves performance—we propose a greedy sampling approach.

In round $t$, let $\mathcal{C}^{t-1}$ be the set of clients selected in round $t-1$. Clients with greater similarity to those in $\mathcal{C}^{t-1}$ will have a higher probability of being selected. Specifically, the sampling probability for client $i$ is calculated as:

$$p_i^t = \frac{\exp(\mathrm{S}(\boldsymbol{\beta}_i, \mathcal{C}^{t-1})/\tau)}{\sum_{j=1}^{N} \exp(\mathrm{S}(\boldsymbol{\beta}_j, \mathcal{C}^{t-1})/\tau)} . \tag{1}$$

Here, $\tau$ is the hyper-parameter controlling the sampling distribution shape, and the similarity function $S$ is defined as:

$$\mathrm{S}(\boldsymbol{\beta}_i, \mathcal{C}^{t-1}) = \frac{1}{2N^{t-1}} \sum_{j=1}^{|\mathcal{C}^{t-1}|} N_j \left( \mathrm{sim}(\boldsymbol{\beta}_i^f, \boldsymbol{\beta}_j^f) + \mathrm{sim}(\boldsymbol{\beta}_i^l, \boldsymbol{\beta}_j^l) \right) ,$$

where $N^{t-1} = \sum_{j \in \mathcal{C}^{t-1}} N_j$, and $N_j$ is the number of samples of client $j$. The "sim" refers to cosine similarity. In practical implementation, to avoid resampling identical clients, those already sampled within $\frac{M}{2|\mathcal{C}^{t-1}|}$ rounds will have $p_i^t$ set to 0, where $M$ is the total number of clients.

**Case study 2: improved model aggregation.** The enhanced model aggregation strategy follows the same idea as client sampling. In detail, we assign higher aggregation weights to clients with greater similarity to those in previous rounds. To achieve this, motivated by the Multiplicative Weight Update algorithm (MWU) (Arora et al., 2012), we define the following optimization problem for deriving the aggregation weights $p_{i,g}^t$.

$$\max_{p_{i,g}^t} \mathcal{L}_{agg} = \underbrace{\sum_{i \in \mathcal{C}^t} p_{i,g}^t \left( \sum_{\tau=1}^t \gamma^{t-\tau} \mathsf{S}(\beta_i, \mathcal{C}^\tau) \right)}_{A_1 := \text{profit function term}} + \underbrace{\lambda_1 \sum_{i \in \mathcal{C}^t} p_{i,g}^t \log \frac{q_i^t}{p_{i,g}^t}}_{A_2 := \text{entropy term}} + \underbrace{\lambda_0 (\sum_{i \in \mathcal{C}^t} p_{i,g}^t - 1)}_{A_3 := \text{regularization term}}, \quad (2)$$

where $\gamma$ is a hyper-parameter controlling the weights of historical information. $A_1$ denotes the profit function in the MWU algorithm, encouraging higher aggregation probability for clients with greater similarity. $A_2$ represents the entropy term, where $q_i^t = N_i/N^t$ denotes the prior distribution of aggregation probability (Li et al., 2020; Balakrishnan et al., 2021). $A_2$ evaluates the risk associated with the aggregation process. $A_3$ serves as a regularization term, ensuring the total aggregation weights sum to 1.

In order to solve (2), we derive the aggregation weights $p_{i,g}^t$ on the communication round $t$ below, and defer the corresponding proof to Appendix A:

$$p_{i,g}^t = \frac{q_i^t \exp\left( \frac{1}{\lambda_1} \sum_{\tau=1}^t \gamma^{t-\tau} \mathsf{S}(\beta_i, \mathcal{C}^\tau) \right)}{\sum_{j \in \mathcal{C}^t} q_j^t \exp\left( \frac{1}{\lambda_1} \sum_{\tau=1}^t \gamma^{t-\tau} \mathsf{S}(\beta_j, \mathcal{C}^\tau) \right)}, \quad (3)$$

where $\lambda_1$ denotes the heat parameter, controlling the entropy term's strength. A higher $\lambda_1$ value emphasizes entropy, resulting in a more evenly distributed set of aggregation weights $p_{i,g}^t$. $\lambda_0$ is tuning-free, as demonstrated in Appendix A, fixing $\sum_{i \in \mathcal{C}^t} p_{i,g}^t = 1$ results in a constant value for $\lambda_0$, which subsequently disappears from the Eq (3).

**Case study 3: improved local training.** According to Definition 3.1, the generated client feature index $\beta_i^f$ contains client-specific information unrelated to label information. Thus, to promote label sensitivity in local features, the local features should be independent of the client feature index $\beta_i^f$. To ensure this, we design the subsequent local objective function to enforce orthogonality between the trained local features $\mathbf{z}_{i,j}$ and the client feature index $\beta_i^f$ for any client $i$.

$$\mathcal{L}(\mathbf{x}, y) = \mathcal{L}_{\text{cls}}(\mathbf{x}, y) + \mathcal{L}_{\text{orth}}(\mathbf{z}_P, \mathbf{B}^f) + \mathcal{L}_{\text{dist}}(\mathbf{z}, \mathbf{z}_P), \quad (4)$$

where $\mathbf{B}^f = [\beta_1^f, \cdots, \beta_N^f]$, $\mathbf{z}$ represents local features, and $\mathbf{z}_P \in \mathbb{R}^{d_i}$ is the projected feature. To handle dimension mismatches between local features $\mathbf{z}_{i,j}$ and client feature indices $\beta_i^f$, we initially project $\mathbf{z} \in \mathbb{R}^d$ to $\mathbf{z}_P \in \mathbb{R}^{d_i}$ using a *trainable* matrix $\mathbf{P} \in \mathbb{R}^{d \times d_i}$ (see Figure 3). An orthogonal loss term $\mathcal{L}_{\text{orth}} = \left\| \mathbf{z}_P \mathbf{B}^f \right\|_1$ further encourages orthogonality between $\mathbf{z}_P$ and $\mathbf{B}^f$.

To preserve maximum information from the original feature $\mathbf{z}$ in $\mathbf{z}_P$, we introduce an additional distillation loss term, denoted as $\mathcal{L}_{dist}$. This term regulates the Kullback-Leibler divergence between the prediction logits of $\mathbf{z}$ and that of $\mathbf{z}_P$.

# 5 EXPERIMENTS

In this section, we investigate if the Client2Vec and the proposed three case studies can improve the FL algorithm performance. More detailed experiment settings and additional experimental results can be found in Appendix D.

## 5.1 EXPERIMENT SETTINGS

**Datasets and models.** In this paper, we utilize three datasets: Shakespeare, CIFAR10, and DomainNet. Shakespeare's partition uses LEAF benchmark's method (Caldas et al., 2018). CIFAR10 is partitioned into 100 clients using Latent Dirichlet Allocation (LDA) (Yurochkin et al., 2019; Hsu et al., 2019) with $\alpha = 0.1$ for label distribution shifts. DomainNet randomly selects 50 classes from the total 345 and divides sub-datasets into 10 clients per domain, resulting in a total of 60 clients. Further dataset partition details are available in Appendix D.2. We randomly select 10% of clients in each of the 100 communication rounds, with a fixed number of 5 local epochs. Additional hyper-parameter details can be found in Appendix D.2.

Table 1: **Performance improvement of Client2Vec.** We assess the performance improvement achieved by employing our proposed three case studies across diverse datasets, neural architectures, and baseline algorithms. Each experiment comprises 100 communication rounds, with the number of local epochs set to 5. We measure the average test accuracy of all clients in each communication round and report the best performance attained across all rounds. The results are then averaged over three seeds. The i indicates improved client sampling, ii indicates the improved model aggregation, and iii indicates the improved local training. The MI indicates the maximum improvement of Client2Vec over baselines. The weight of the local regularization term in iii is set to 1.0 for FEDERATED strategy, and 5.0 for GLOBAL strategy.

| Datasets | Algorithms | Original | Client2Vec (FEDERATED) | | | Client2Vec (GLOBAL) | | | MI |
|---|---|---|---|---|---|---|---|---|---|
| | | - | + i | + i+ ii | + i+ ii+ iii | + i | + i+ ii | + i+ ii+ iii | - |
| Shakespeare (LSTM) | FedAvg | 49.93 ±0.12 | 50.33 ±0.03 | 50.28 ±0.04 | **50.51** ±0.10 | 50.30 ±0.08 | 50.38 ±0.07 | 50.40 ±0.08 | 0.58 |
| | FedAvgM | 49.97 ±0.09 | 50.29 ±0.01 | 50.24 ±0.01 | 50.43 ±0.01 | 50.29 ±0.19 | 50.24 ±0.03 | **50.60** ±0.22 | 0.63 |
| | FedDyn | 50.23 ±0.08 | 50.47 ±0.17 | 50.43 ±0.14 | 50.55 ±0.02 | 50.64 ±0.13 | 50.49 ±0.19 | **50.71** ±0.11 | 0.48 |
| | Moon | 50.09 ±0.08 | 50.35 ±0.02 | 50.35 ±0.16 | **50.54** ±0.03 | 50.36 ±0.01 | 50.38 ±0.11 | 50.52 ±0.22 | 0.45 |
| | FedLC | 49.89 ±0.19 | 50.43 ±0.08 | 50.37 ±0.09 | 50.46 ±0.05 | 50.34 ±0.05 | 50.29 ±0.07 | **50.50** ±0.05 | 0.61 |
| CIFAR10 (ResNet18) | FedAvg | 42.24 ±2.18 | 44.60 ±0.74 | 44.10 ±0.20 | **59.29** ±2.58 | 45.56 ±0.18 | 46.49 ±0.08 | 58.28 ±4.95 | 17.05 |
| | FedAvgM | 42.56 ±2.23 | 45.81 ±1.36 | 45.05 ±1.24 | 63.48 ±2.16 | 46.55 ±0.83 | 46.24 ±1.36 | **69.37** ±4.49 | 26.81 |
| | FedDyn | 37.22 ±3.26 | 39.49 ±0.01 | 39.45 ±0.20 | 69.10 ±1.17 | 39.42 ±0.08 | 39.84 ±0.16 | **70.59** ±3.86 | 33.37 |
| | Moon | 41.12 ±1.23 | 44.28 ±0.45 | 43.79 ±0.43 | 60.26 ±3.29 | 45.20 ±0.36 | 44.85 ±1.22 | **65.55** ±0.27 | 24.43 |
| | FedLC | 29.31 ±0.01 | 29.62 ±0.13 | 30.65 ±0.29 | **42.20** ±2.14 | 31.04 ±0.98 | 30.37 ±0.82 | 40.27 ±1.00 | 12.89 |
| DomainNet (MobileNet V2) | FedAvg | 46.31 ±1.36 | 50.78 ±1.42 | 53.83 ±0.37 | 56.43 ±3.08 | 52.37 ±0.59 | 54.67 ±0.77 | **57.43** ±0.13 | 11.12 |
| | FedAvgM | 45.50 ±1.21 | 50.61 ±1.73 | 55.70 ±0.69 | **58.34** ±0.01 | 53.50 ±2.33 | 53.56 ±0.81 | 57.44 ±1.04 | 12.84 |
| | FedDyn | 45.41 ±0.89 | 47.24 ±0.29 | 49.90 ±0.34 | **55.53** ±1.49 | 52.42 ±0.23 | 50.68 ±0.26 | 53.33 ±0.26 | 10.12 |
| | Moon | 50.56 ±0.89 | 59.39 ±0.47 | 59.54 ±0.44 | 57.03 ±0.60 | **60.48** ±0.10 | 59.93 ±0.41 | 57.50 ±0.52 | 9.92 |
| | FedLC | 45.48 ±3.59 | 50.40 ±0.43 | 51.27 ±0.66 | **57.92** ±0.67 | 54.60 ±1.45 | 56.34 ±2.78 | 57.41 ±0.06 | 12.44 |
| | FedIIR | 49.32 ±0.84 | 48.11 ±0.18 | 50.28 ±1.10 | 52.74 ±1.07 | **57.05** ±1.84 | 53.74 ±0.27 | 51.86 ±1.08 | 7.73 |

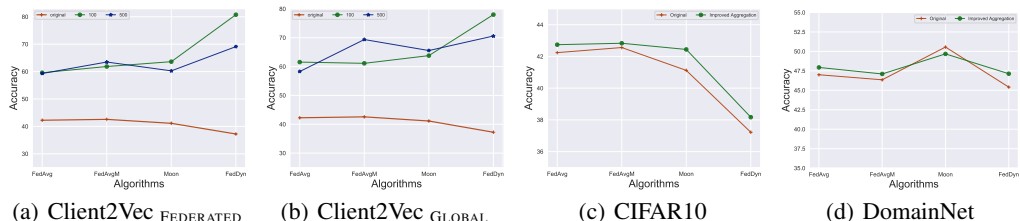

(a) Client2Vec FEDERATED    (b) Client2Vec GLOBAL    (c) CIFAR10    (d) DomainNet

Figure 4: **Ablation studies on the number of training epochs and improved model aggregation for Client2Vec.** 'Original' represents the algorithms in their original form, without enhancements, while other results consider all three case studies with varying epoch numbers. The Figures 4(c) and 4(d) utilize client indices generated by the GLOBAL strategies.

**Baseline Algorithms.** We have selected widely recognized FL baselines for our study, encompassing established methods such as FedAvg (McMahan et al., 2016), FedAvgM (Hsu et al., 2019), Fed-Dyn (Acar et al., 2021), and Moon (Li et al., 2021). Additionally, we have incorporated more recently introduced baselines, namely FedLC (Zhang et al., 2022) and FedIIR (Guo et al., 2023a). It is important to note that FedIIR is specifically tailored for addressing feature shift tasks and, as a result, its evaluation is limited to the DomainNet dataset.

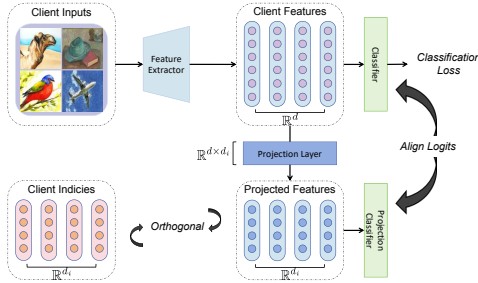

Figure 3: **Workflow of the improved local training (case study 3).** The projection layer is to project client features to the same dimension with client feature index $\beta_i^f$, and the projection classifier is to ensure the projected features and the original client features contain similar information.

## 5.2 NUMERICAL RESULTS

**Superior performance of Client2Vec on all three case studies.** In Table 1, we evaluate Client2Vec's performance in three case studies: enhanced client sampling, improved model aggregation, and refined local training. The results reveal the following insights: (1) *Each case study shows performance improvements across all baselines*, highlighting the potential of generated client indices to enhance FL algorithms. (2) *Enhanced local training provides the most significant performance boost,*

Table 2: **Ablation studies on Client2Vec.** We present naive baselines for Client2Vec. Feature Average: We compute the average of CLIP image features for all data samples within each client as the client index. Class Prototypes: We form each client's index by averaging CLIP image features for each class and concatenating the resulting class prototypes. Local Model Weights: We use the classifier model weights at the end of each training epoch as the client index for each client. Notably, unlike other methods, these local model weights change with each epoch, posing challenges for direct integration into our sampling and aggregation mechanisms.

| CIFAR10 | Sampling | Sampling + Aggregation | Sampling + Aggregation + Local Training |
|---|---|---|---|
| *Ablation Study on Types of Client Index* | | | |
| Feature Average | 52.95 | 31.99 | 11.40 |
| Class Prototypes | 44.89 | 49.88 | 15.41 |
| Local Model Weights | - | - | 32.85 |
| *Ablation Study on Generating Client Index* | | | |
| Omit Orthogonal Loss | 38.58 | 44.39 | 15.32 |
| Omit Text Align Loss | 44.91 | 43.18 | 54.35 |
| Omit Reconstruction Loss | 43.19 | 30.86 | 31.65 |
| *Ablation Study on Label and Feature Index* | | | |
| Utilize Only Feature Index | 42.34 | 23.57 | 44.99 |
| Utilize Only Label Index | 45.99 | 46.48 | |
| *Ablation Study on Local Training* | | | |
| Omit Orthogonal Regularization in Improved Local Training | - | - | 43.71 |
| Client2Vec | 45.56 | 46.49 | 58.28 |

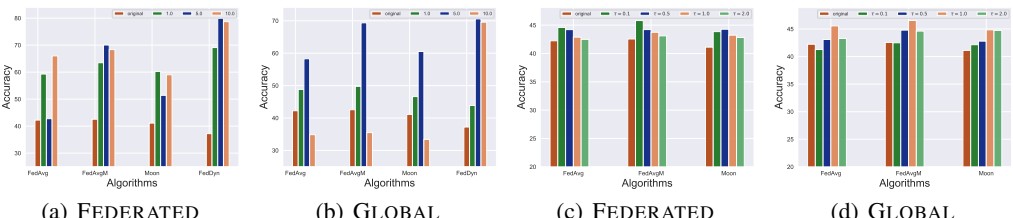

(a) FEDERATED  (b) GLOBAL  (c) FEDERATED  (d) GLOBAL

Figure 5: **Ablation studies on improved local training and improved client sampling.** We use the CIFAR10 dataset and client indices from both FEDERATED and GLOBAL strategies. Figure 5(a) and 5(b) different weights for Eq (4); Figure 5(c) and 5(d) vary hyperparameter $\tau$ in Eq (1).

Table 3: **Simulation time comparison.** We compare the simulation time of Client2Vec and FedAvg on DomainNet dataset.

| | Generate Client Index | Training Total | Training achieve FedAvg best performance | Total achieve FedAvg best performance |
|---|---|---|---|---|
| Client2Vec | 632s | 24583s | 10817s | 11449s |
| FedAvg | 0s | 26224s | 26224s | 26224s |

emphasizing the importance of refining local features for addressing distribution shifts. (3) *The* FEDERATED *strategy consistently matches the* GLOBAL *strategy in performance*, except for improved client sampling, where the GLOBAL strategy surpasses, showcasing its superior capability in assessing client similarities (see Figure 2). (4) The performance gain from improved model aggregation seems somewhat random compared to other case studies. This might be due to the shared intuition between improved client sampling and model aggregation, limiting further improvements when combining these approaches. However, *solely using improved model aggregation consistently outperforms the original algorithms*, as seen in Figure 4(c) and 4(d).

*In summary, utilizing the client indices significantly boost the model performance. In practice, we can select which case studies to use, and we recommend combining all three case studies as this approach provides a stable and significant performance gain compared to the baseline algorithms.*

**All the components of Client2Vec are necessary.** In Table 2, we conduct ablation studies on Client2Vec. The results indicate that: (1) *The original Client2Vec achieves the highest final performance among different client vector candidates.* Although feature average and class prototypes show potential for aiding in sampling, they cannot be readily employed in our local training phase, which significantly contributes to the effectiveness of the Client2Vec algorithm. (2) *All three losses are essential for Client2Vec's effectiveness.* The orthogonal loss notably enhances the local training phase, while removing the text align loss and reconstruction loss significantly diminishes model performance. (3) *Both feature and label indices are vital for optimizing Client2Vec's performance.* The absence of the feature index hinders improved local training. Additionally, since the CIFAR10

Table 4: **Performance of Client2Vec on various network architectures.** We evaluate the performance of Client2Vec on the DomainNet dataset using diverse network architectures. The term 'Original' refers to the initial form of the algorithms, while Client2Vec (FEDERATED) and Client2Vec (GLOBAL) applied all three case studies. Each experiment involves 100 communication rounds, with the number of local epochs set to 5. We gauge the average test accuracy of all clients in each communication round and report the highest performance achieved across all rounds. The results are averaged over three seeds. For the VIT experiments, we use the CCT-7/3x1 models (Hassani et al., 2021).

| DomainNet | MobileNet V2 (Pre-Trained) | | | ResNet18 (Pre-Trained) | | | VIT (From Scratch) | | |
|---|---|---|---|---|---|---|---|---|---|
| | Original | Client2Vec | | Original | Client2Vec | | Original | Client2Vec | |
| | | FEDERATED | GLOBAL | | FEDERATED | GLOBAL | | FEDERATED | GLOBAL |
| FedAvg | 46.31 ±1.36 | 56.43 ±3.08 | 57.43 ±0.13 | 56.66 ±0.50 | 61.27 ±0.05 | 60.95 ±0.09 | 33.09 ±0.01 | 33.50 ±0.20 | 33.86 ±0.02 |
| FedAvgM | 45.50 ±1.21 | 58.34 ±0.01 | 57.44 ±1.04 | 57.44 ±0.42 | 61.22 ±0.11 | 60.81 ±0.18 | 33.67 ±0.56 | 34.47 ±0.20 | 34.21 ±0.11 |
| FedDyn | 45.41 ±0.89 | 51.49 ±0.17 | 53.33 ±0.26 | 58.17 ±0.61 | 61.67 ±0.42 | 59.88 ±0.42 | 29.57 ±0.40 | 31.64 ±0.13 | 31.36 ±0.12 |
| MOON | 50.56 ±0.89 | 57.03 ±0.60 | 57.50 ±0.52 | 53.80 ±0.46 | 60.76 ±0.25 | 59.90 ±0.17 | 32.29 ±0.52 | 33.58 ±0.12 | 33.73 ±0.03 |

dataset is partitioned using the Dirichlet method, introducing label shifts instead of feature shifts, the label index is crucial for enhancing sampling and model aggregation performance in this context. (4) *Orthogonal loss is crucial for achieving optimal performance in local training.* Without it, local training fails to surpass the performance of the original FedAvg.

**Ablation studies on the number of training epochs for Client2Vec.** We conduct ablation studies on the Client2Vec algorithm, varying the number of training epochs as shown in Figure 4(a) and 4(b). The results indicate that: (1) Client indices generated with 100 or 500 training epochs notably enhance FL algorithm performance. (2) Increasing the number of training epochs for Client2Vec does not consistently lead to better results, as 100 epochs achieve similar performance to 500 epochs in most cases.

**Ablation studies on hyper-parameters of improved client sampling.** In Figure 5(c) and 5(d), we perform ablation studies on the heat parameter $\tau$ in Eq (1). The results indicate that (1) algorithms with improved client sampling consistently outperform the original algorithms across various $\tau$ values; (2) the optimal $\tau$ value is smaller for client indices trained using the FEDERATED strategy compared to the GLOBAL strategy. This observation aligns with our previous findings that GLOBAL strategy-trained client indices exhibit larger inter-client distances.

**Ablation studies on hyper-parameters of improved local training.** In Figure 5, ablation studies on the weight of the local regularization term (Eq (4)) were conducted. The findings suggest that: (1) Using weights of $1.0$ for the FEDERATED strategy and $5.0$ for the GLOBAL strategy yields favorable results for all algorithms. (2) FedDyn exhibits higher resilience to changes in the weights of the local regularization terms.

**Computation time comparison.** In Table 3, we show the simulation time of Client2Vec and FedAvg. Results show that (1) The additional computational overhead for generating the client index is relatively insignificant compared to the subsequent training stage; (2) From the 'Total achieve FedAvg best performance' column, Client2Vec requires less computational time to achieve comparable performance to FedAvg, particularly noticeable on larger-scale datasets such as DomainNet.

**Ablation Studies on Various Model Architectures.** In Table 8, we show how Client2Vec improves performance with different model architectures. Our results reveal that: (1) Client2Vec significantly boosts the performance of original algorithms in all settings, and (2) pre-trained models like MobileNet V2 and ResNet18 produce better results, while Client2Vec also enhances the performance of VIT models trained from scratch.

# 6 CONCLUSION AND FUTURE WORKS

In this paper, we explore the potential of enhancing FL algorithm performance through client index vectors. Our three case studies clearly demonstrate the significant improvement in FL algorithm performance achieved through client indices, highlighting client indexing as a valuable avenue for FL algorithm enhancement. It's important to note that these case studies may not cover all FL training scenarios. Investigating the impact of client indices on other aspects, such as personalization and clustering, would be valuable.

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

CONTENTS OF APPENDIX

## A  PROOF OF AGGREGATION WEIGHTS

**Theorem A.1** (Aggregation weights)**.** *Define the following objective function*

$$\max_{p_{i,g}^t} \mathcal{L}_{agg} = \sum_{i \in \mathcal{S}^t} p_{i,g}^t \left( \sum_{\tau=1}^t \gamma^{t-\tau} S(\beta_i, \mathcal{S}^\tau) \right) + \lambda_1 \sum_{i \in \mathcal{S}^t} p_{i,g}^t \log \frac{q_i^t}{p_{i,g}^t} + \lambda_0 (\sum_{i \in \mathcal{S}^t} p_{i,g}^t - 1), \quad (5)$$

*where $p_{i,g}^t$ is the aggregation weights on communication round $t$, $S$ is the similarity function, and $q_i^t$ is a prior distribution. Solving this optimization problem, the optimal $p_{i,g}^t$ is given by*

$$p_{i,g}^t = \frac{q_i^t \exp\left( \frac{1}{\lambda_1} \sum_{\tau=1}^t \gamma^{t-\tau} dist(\beta_i, \mathcal{S}^\tau) \right)}{\sum_{j \in \mathcal{S}^t} q_j^t \exp\left( \frac{1}{\lambda_1} \sum_{\tau=1}^t \gamma^{t-\tau} dist(\beta_j, \mathcal{S}^\tau) \right)} . \quad (6)$$

*Proof.* Taking the derivation, we have

$$\frac{\partial \mathcal{L}_{agg}}{\partial p_{i,g}^t} = \sum_{\tau=1}^t \gamma^{t-\tau} \text{dist}(\beta_i, \mathcal{S}^\tau) + \lambda_1 \left( \log q_i^t - \log p_{i,g}^t - 1 \right) + \lambda_0 , \quad (7)$$

then we have

$$p_{i,g}^t = \exp\left( \frac{1}{\lambda_1} \sum_{\tau=1}^t \gamma^{t-\tau} \text{dist}(\beta_i, \mathcal{S}^\tau) + \log q_i^t - 1 + \frac{\lambda_0}{\lambda_1} \right) . \quad (8)$$

Because $\sum_{i \in \mathcal{S}^t} p_{i,g}^t = 1$, we have

$$1 - \frac{\lambda_0}{\lambda_1} = \log\left( \sum_{i \in \mathcal{S}^t} \exp\left( \frac{1}{\lambda_1} \sum_{\tau=1}^t \gamma^{t-\tau} \text{dist}(\beta_i, \mathcal{S}^\tau) + \log q_i^t \right) \right) \quad (9)$$

$$= \log\left( \sum_{i \in \mathcal{S}^t} q_i^t \exp\left( \frac{1}{\lambda_1} \sum_{\tau=1}^t \gamma^{t-\tau} \text{dist}(\beta_i, \mathcal{S}^\tau) \right) \right) , \quad (10)$$

Then combine Equations (8) and (10) we have

$$p_{i,g}^t = \frac{q_i^t \exp\left( \frac{1}{\lambda_1} \sum_{\tau=1}^t \gamma^{t-\tau} \text{dist}(\beta_i, \mathcal{S}^\tau) \right)}{\sum_{j \in \mathcal{S}^t} q_j^t \exp\left( \frac{1}{\lambda_1} \sum_{\tau=1}^t \gamma^{t-\tau} \text{dist}(\beta_j, \mathcal{S}^\tau) \right)} \quad (11)$$

$\square$

## B    RELATED WORKS

**Distribution shifts in FL.**    Federated Learning (FL) is introduced as a methodology for training machine learning models in a distributed manner, wherein local data is retained and not exchanged between the central server and individual clients. FedAvg (McMahan et al., 2016; Lin et al., 2020b), serving as a foundational algorithm in this domain, advocates the use of local Stochastic Gradient Descent (local SGD) to alleviate the communication burden. Nevertheless, the performance of FL algorithms is substantially impeded by distribution shifts among clients. Addressing these local distribution shifts has emerged as a primary focus in FL research (Li et al., 2020; Karimireddy et al., 2020b;a; Guo et al., 2023b; Jiang & Lin, 2023). Many existing works address label distribution shifts by incorporating additional regularization terms (Li et al., 2020; Karimireddy et al., 2020b; Guo et al., 2021; Lee et al., 2022; Mendieta et al., 2022), enhancing feature learning (Tang et al., 2022; Shi et al., 2023; Li et al., 2021; Zhou et al., 2023), and improving classifiers (Luo et al., 2021; Li et al., 2023c). Regarding feature distribution shifts, the majority of FL methods concentrate on the out-of-domain generalization problem. This objective aims to train robust models capable of generalizing to previously unseen feature distributions (Nguyen et al., 2022; Li et al., 2023b; Guo et al., 2023a). Approaches include investigating special cases (Reisizadeh et al., 2020), integrating domain generalization algorithms in FL scenarios, such as domain-robust optimization (Mohri et al., 2019; Deng et al., 2021), and training domain-invariant features (Peng et al., 2019b; Wang et al., 2022; Shen et al., 2021; Sun et al., 2022; Gan et al., 2021). Notably, recent research has also considered concept shifts by leveraging clustering methods (Jothimurugesan et al., 2022; **?**; Guo et al., 2023c). In this study, we address the challenge of distribution shifts in FL from another perspective—enhancing the performance of FL algorithms prior to the training stage. Our approach holds the potential for seamless integration with the aforementioned algorithms, and consider both feature and label distribution shifts.

**Information sharing in FL.**    Various methods have been developed to address the challenge of distribution shifts among clients (Zhao et al., 2018; Jeong et al., 2018; Long et al., 2021). One approach involves the sharing of information among clients, such as the exchange of local distribution statistics (Shin et al., 2020; Zhou & Konukoglu, 2022), data representations (Hao et al., 2021; Tan et al., 2022), and prediction logits (Chang et al., 2019; Luo et al., 2021). Additionally, techniques leveraging global proxy datasets have been introduced to enhance FL training (Lin et al., 2020a; Duan et al., 2019). Notably, FedMix (Yoon et al., 2020) and FedBR (Guo et al., 2023b) generate privacy-protected augmentation data by averaging local batches, subsequently improving the local training process. VHL (Tang et al., 2022) employs randomly initialized generative models to produce virtual data, compelling local features to closely align with those of same-class virtual data. FedFed (Yang et al., 2023) proposes a dataset distillation method, amalgamating distilled datasets into all clients' local datasets to mitigate distribution shifts. In comparison to existing approaches, Client2Vec presents several advantages: (1) the index generation process is decoupled from the FL training process, thereby avoiding any additional burden on FL training; (2) Client2Vec generates only one index vector per client, enhancing efficiency; (3) Client2Vec contributes to the whole FL training stage, encompassing client sampling, model aggregation, and local training processes.

## C    PRELIMINARIES

In this section, we present essential background information on the techniques and definitions employed in this paper to facilitate comprehension.

### C.1    DOMAIN INDEXING

The Domain Generalization (DG) tasks are designed to address the cross-domain generalization problem by generating domain-invariant features. Typically, DG methods aim to establish independence between a data point's latent representation and its domain identity, represented by a one-hot vector indicating the source domain (Ganin et al., 2016; Tzeng et al., 2017; Zhao et al., 2017). However, recent studies have demonstrated that utilizing a domain index, which is a real-value scalar (or vector) embedding domain semantics, as a substitute for domain identity, significantly enhances domain generalization performance (Wang et al., 2020; Xu et al., 2021).

For example, in the work by Wang et al. (2020), sleeping stage prediction models were adapted across patients with varying ages, using "age" as the domain index. This approach yielded superior performance compared to traditional models that categorized patients into groups based on age, employing discrete group IDs as domain identities.

Nevertheless, obtaining domain indices may not always be feasible in practical scenarios. To overcome this challenge, Xu et al. (2022) formally defined the domain index and introduced variational domain indexing (VDI) to infer domain indices without prior knowledge. The definition of the domain index in (Xu et al., 2022) is illustrated as follows.

**Definition of domain index.** Consider the unsupervised domain adaptation setting involving a total of $N$ domains, each characterized by a domain identity $k \in \mathcal{K} = [N] \triangleq \{1, \ldots, N\}$. Here, $k$ belongs to either the source domain identity set $\mathcal{K}_s$ or the target domain identity set $\mathcal{K}_t$. Every domain $k$ comprises $D_k$ data points. The task involves $n$ labeled data points $\{(\mathbf{x}i^s, y_i^s, k_i^s)\}_{i=1}^n$ originating from source domains ($k_i^s \in \mathcal{K}_s$) and $m$ unlabeled data points $\{\mathbf{x}i^t, k_i^t\}_{i=1}^m$ from target domains ($k_i^t \in \mathcal{K}_t$). The objectives are twofold: (1) predict the labels $\{y_i^t\}_{i=1}^m$ for the target domain data, and (2) deduce global domain indices $\boldsymbol{\beta}_k \in \mathbb{R}^{B_\beta}$ for each domain and local domain indices $\mathbf{u}_i \in \mathbb{R}^{B_u}$ for each data point. It is important to note that each domain possesses a single global domain index but multiple local domain indices, with one corresponding to each data point in the domain. The data encoding generated from an encoder that takes $\mathbf{x}$ as input is represented as $\mathbf{z} \in \mathbb{R}^{B_z}$. The mutual information is denoted by $I(\cdot; \cdot)$.

**Definition C.1** (Domain Index). *Given data $\mathbf{x}$ and label $y$, a domain-level variable $\boldsymbol{\beta}$ and a data-level variable $\mathbf{u}$ are called global and local domain indices, respectively, if there exists a data encoding $\mathbf{z}$ such that the following holds:*

- *Independence between $\boldsymbol{\beta}$ and $\mathbf{z}$: Global domain index $\beta$ is independent of data encoding $\mathbf{z}$, i.e., $\boldsymbol{\beta} \perp\!\!\!\perp \mathbf{z}$, or equivalently $I(\boldsymbol{\beta}; \mathbf{z}) = 0$. This is to encourage domain-invariant data encoding $\mathbf{z}$.*

- *Information Preservation of $\mathbf{z}$: Data encoding $\mathbf{z}$, local domain index $\mathbf{u}$, and global domain index $\boldsymbol{\beta}$ preserves as much information on $\mathbf{x}$ as possible, i.e., maximizing $I(\mathbf{x}; \mathbf{u}, \boldsymbol{\beta}, \mathbf{z})$. This is to prevent $\boldsymbol{\beta}$ and $\mathbf{u}$ from collapsing to trivial solutions.*

- *Label Sensitivity of $\mathbf{z}$: The data encoding $\mathbf{z}$ should contain as much information on the label $y$ as possible to maximize prediction power, i.e., maximizing $I(y; \mathbf{z})$ conditioned on $\mathbf{z} \perp\!\!\!\perp \boldsymbol{\beta}$. This is to make sure the previous two constraints on $\boldsymbol{\beta}$, $\mathbf{u}$, and $\mathbf{z}$ do not harm prediction performance.*

In this paper, we extend the Definition C.1 to Definition 3.1 by incorporating both client feature index and client label index.

## C.2 CLIP

CLIP (Radford et al., 2021) is a cross-modal model that establishes a connection between vision and natural language by projecting image and text embeddings onto a shared space. When presented with an image $\mathbf{I}$ and a corresponding descriptive sentence denoted as $\mathbf{T}$, the CLIP image encoder and text encoder encode the image and text into image embedding $\mathbf{D}$ and text embedding $\mathbf{L}$, respectively. Subsequently, the embeddings $\mathbf{D}$ and $\mathbf{L}$ are aligned to achieve a large cosine similarity, thereby harmonizing the vision and language embedding spaces.

# D    ADDITIONAL EXPERIMENT RESULTS

## D.1    WORKFLOW OF CLIENT2VEC ON LANGUAGE DATASETS

In Figure 6, we depict the workflow of Client2Vec on language datasets. The primary distinction between Figure 1 and Figure 6 arises from the methods employed for encoding data and labels. Specifically, for language datasets, particularly in the context of the next character prediction task, the data is encoded as "The next character of {data}", while the label is encoded as "Character {label}".

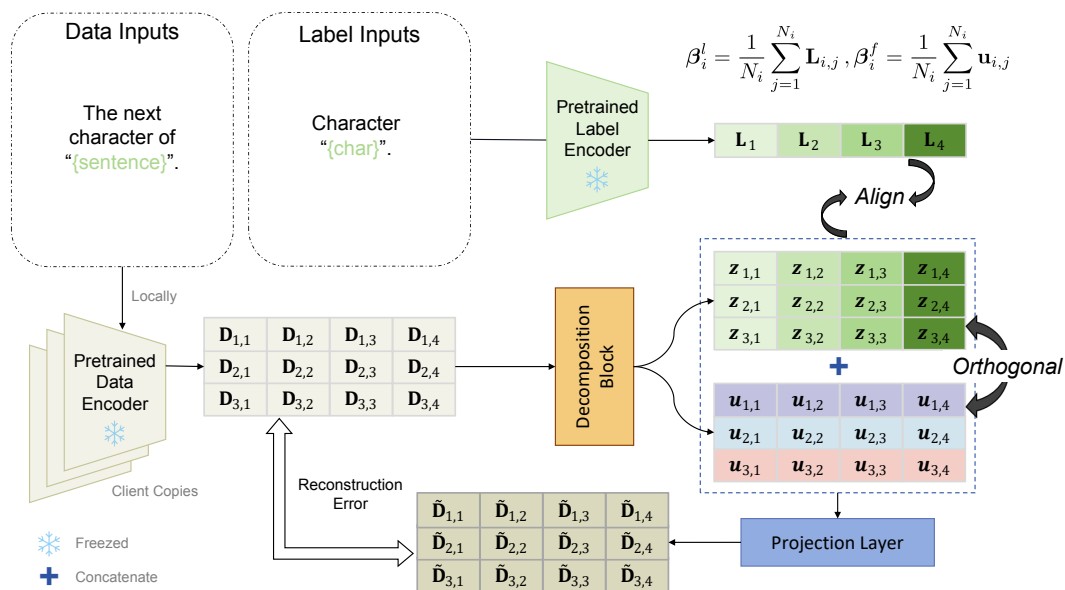

Figure 6: **Overview of the workflow of the Client2Vec on language datasets.**

In both cases, the CLIP text encoder is utilized by both the data encoder and label encoder for this task.

## D.2 EXPERIMENT SETTINGS

**Dataset partition.** The dataset partition follows the widely used settings in FL. In detail, we consider three datasets in this paper, and the details are listed as the follows.

- **Shakespeare:** The partition of Shakespeare dataset directly use the partition method provided by LEAF benchmark (Caldas et al., 2018), and we set the fraction of data sample to 0.1, fraction of data in training set is set to 0.8, and minimum number of samples per user is set to 40.
- **CIFAR10:** We use the Latent Dirichlet Allocation (LDA) (Yurochkin et al., 2019; Hsu et al., 2019) method with parameter $\alpha = 0.1$ to introduce label distribution shifts among clients. The dataset is partitioned into 100 clients.
- **DomainNet:** We randomly choose 50 classes from the overall 345 classes from DomainNet dataset. Sub-datasets of each domain are partitioned into 10 clients, resulting in 60 clients in total. Images are resized to $64 \times 64$.

**Training details and hyper-parameters.** For every dataset and algorithm, we randomly select 10% of clients in each communication round and execute a total of 100 communication rounds. We employ the SGD optimizer, with a momentum setting of 0.9 for the DomainNet dataset, and a weight decay set to 5e-5. The number of local epochs is fixed at 5, and the learning rate is set to 1e-2. The experiments are conduct on single NVIDIA 3090 GPU. The hyperparameters for our enhanced case studies are detailed below.

- **Improved client sampling.** The heat parameter $\tau$ in Eq (1) is tuned in $[0.1, 0.5, 1.0, 2.0]$.
- **Improved model aggregation.** We choose the optimal results by choosing $\gamma = [0.1, 0.5, 0.9]$, and set $\lambda_1 = 1.0$ by default in Eq (8).
- **Improved local training.** For algorithms without extra local regularization terms, such as FedAvg, FedAvgM, and FedLC, the weights assigned to $\mathcal{L}_{orth}$ and $\mathcal{L}_{dist}$ are explicitly fixed at 1.0. In contrast, for approaches incorporating additional local regularization terms, such as Moon, FedDyn, and FedIIR, the weights assigned to $\mathcal{L}_{orth}$ and $\mathcal{L}_{dist}$ are set equal to the respective values of those additional local regularization terms in the respective algorithms.

The hyper-parameters utilized for each baseline algorithms are listed below.

- **FedAvgM:** The server momentum is tuned in $[0.1, 0.5, 1.0]$.

Table 5: **Ablation studies on improved client sampling.** We conduct ablation studies on hyper-parameter $\tau$ in Equation (1). The term 'Original' refers to the algorithm in its initial form, where the improved client sampling is not applied. This ablation study focuses on improved client sampling, without integrating the other case studies involving enhanced model aggregation and improved local training.

| CIFAR10 | Original | Client2Vec (Federated) | | | | Client2Vec (Global) | | | |
|---|---|---|---|---|---|---|---|---|---|
| | - | $\tau = 0.1$ | $\tau = 0.5$ | $\tau = 1.0$ | $\tau = 2.0$ | $\tau = 0.1$ | $\tau = 0.5$ | $\tau = 1.0$ | $\tau = 2.0$ |
| FedAvg | 42.24 | 44.60 | 44.21 | 42.88 | 42.49 | 41.28 | 43.10 | **45.56** | 43.28 |
| FedAvgM | 42.56 | 45.81 | 44.22 | 43.74 | 43.11 | 42.50 | 44.80 | **46.55** | 44.62 |
| Moon | 41.12 | 43.86 | 44.28 | 43.23 | 42.82 | 42.15 | 42.80 | **44.85** | 44.74 |

Table 6: **Ablation studies on training epochs of Client2Vec.** We perform ablation studies on the training epochs of DSA-IGN, incorporating all three case studies.

| CIFAR10 | Original | Client2Vec (Federated) | | Client2Vec (Global) | |
|---|---|---|---|---|---|
| | - | $E = 100$ | $E = 500$ | $E = 100$ | $E = 500$ |
| FedAvg | 42.24 | 59.58 | 59.29 | **61.55** | 58.28 |
| FedAvgM | 42.56 | 61.84 | 63.48 | 61.12 | **69.37** |
| Moon | 41.12 | 63.61 | 60.26 | 63.79 | **65.55** |
| FedDyn | 37.22 | **80.75** | 69.10 | 78.01 | 70.59 |

- **FedDyn:** We set $\alpha = 0.1$, and the max gradient norm to 10.
- **Moon:** The heat parameter is set to 0.5, and the weights of local regularization term is tuned in $[0.01, 0.1, 1.0]$.
- **FedLC:** We set $\tau = 1.0$.
- **FedIIR:** We tuned $ema = [0.95, 0.5, 0.1]$, and the weights of local regularization term are set to $1e - 3$.

**Model architectures and training details of DSA-IGN.** The projection layer utilizes a three-layer transformer encoder. Each transformer encoder layer consists of 8 attention heads, with the model dimension set to 32, and the feed-forward layer dimension set to 2048. The projection layer is represented as a matrix with dimensions $1024 \times 512$. Given a batch of CLIP embeddings $\mathbf{D} \in \mathbb{R}^{N \times 512}$, the input for the decomposition block is constructed as $\mathbf{I}_{i,j} = [\mathbf{D}, \mathbf{D}] \in \mathbb{R}^{N \times 1024}$. Subsequently, $\mathbf{I}$ is reshaped into $\tilde{\mathbf{I}} = (N \times 32 \times 32)$, indicating that each sample comprises 32 patches, and each patch has a dimension of 32.

The reshaped $\tilde{\mathbf{I}}$ is fed into the decomposition block, producing an output $\tilde{\mathbf{O}} \in (N \times 32 \times 32)$, which is then reshaped to $\mathbf{O} = (N \times 1024) = [\mathbf{Z}, \mathbf{U}]$. Here, $\mathbf{Z} \in \mathbb{R}^{N \times 512}$ represents the data encoding $\mathbf{z}$ as defined in Definition 3.1, and $\mathbf{U} \in \mathbb{R}^{N \times 512}$ corresponds to the sample feature index $\mathbf{u}$. The input to the projection layer is identical to the output of the decomposition block, represented as $\mathbf{O}$.

### D.3 ABLATION STUDIES ON CLIENT INDEX GENERATION

**Generating client index w/o the use of the diversity loss $\mathcal{L}_{div}$.** As shown in Figure 7, the client feature index $\beta_i^f$ become close to identical when do not use the diversity loss. This result suggest the necessity of using the diversity loss to obtain the meaningful results.

**Using different projection layers in DSA-IGN.** In Figure 8, we use single Linear layer and two-layer MLP as projection layers in DSA-IGN. Results show that both architectures can obtain sufficient meaningful results.

### D.4 ABLATION STUDIES ON CASE STUDIES

In Tables 5, 6, and 7, we conduct ablation studies on the three case studies we introduced in Section 4.

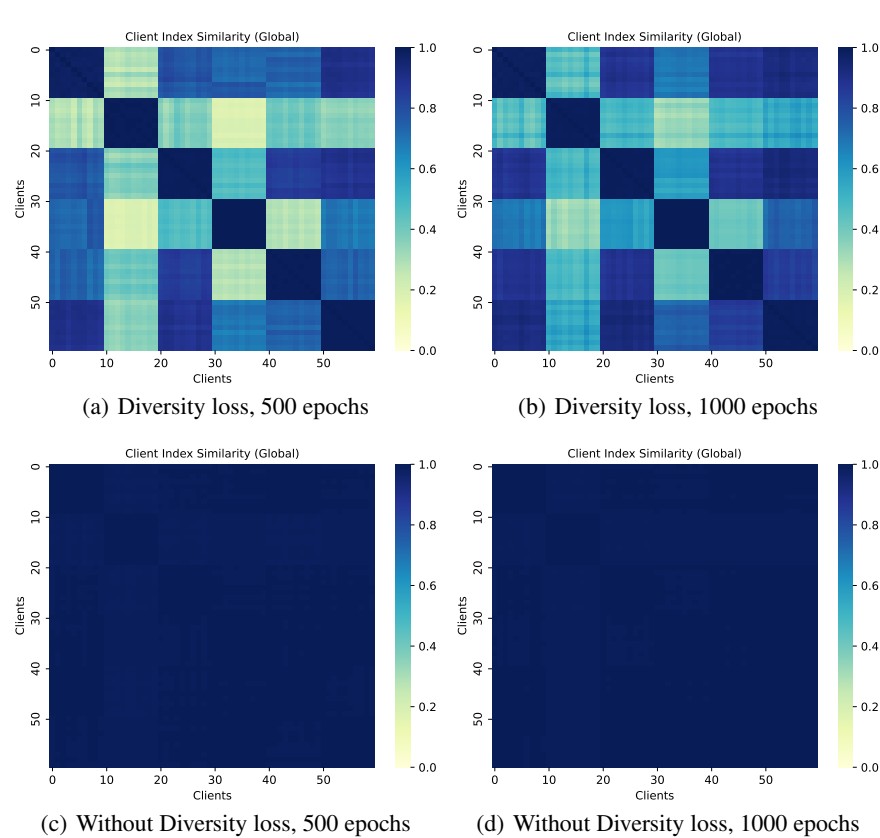

(a) Diversity loss, 500 epochs    (b) Diversity loss, 1000 epochs

(c) Without Diversity loss, 500 epochs    (d) Without Diversity loss, 1000 epochs

Figure 7: **Comparison between client indexed generated with/without diversity loss.** We use the DomainNet dataset with 60 clients, and use the *Global* training strategy. The DSA-IGN is trained by 500 and 1000 global epochs. We resport the cos-similarities of the client feature index $\beta_i^f$.

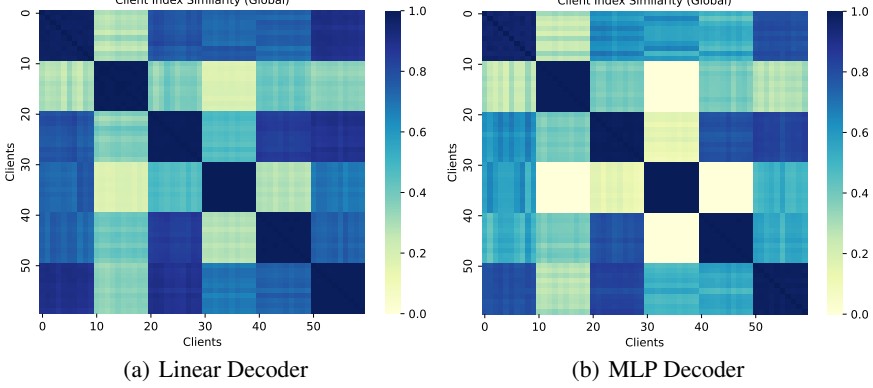

(a) Linear Decoder    (b) MLP Decoder

Figure 8: **Comparison between client indexed generated using different projection layers.** We use the DomainNet dataset with 60 clients, and use the *Global* training strategy. The DSA-IGN is trained by 500 global epochs. We resport the cos-similarities of the client feature index $\beta_i^f$.

Table 7: **Ablation studies on improved local training.** We conduct ablation studies on the weights of the improved local training. All three case studies are incorporated in this setting.

| CIFAR10 | Original | Client2Vec (Federated) | | | Client2Vec (Global) | | |
|---|---|---|---|---|---|---|---|
| | - | 1.0 | 5.0 | 10.0 | 1.0 | 5.0 | 10.0 |
| FedAvg | 42.24 | 59.29 | 42.76 | **66.02** | 48.83 | 58.28 | 34.86 |
| FedAvgM | 42.56 | 63.48 | **70.04** | 68.34 | 49.77 | 69.37 | 35.51 |
| Moon | 41.12 | 60.25 | 51.41 | 59.02 | 46.61 | **60.53** | 33.39 |
| FedDyn | 37.22 | 69.10 | **79.96** | 78.70 | 43.87 | 70.59 | 69.57 |

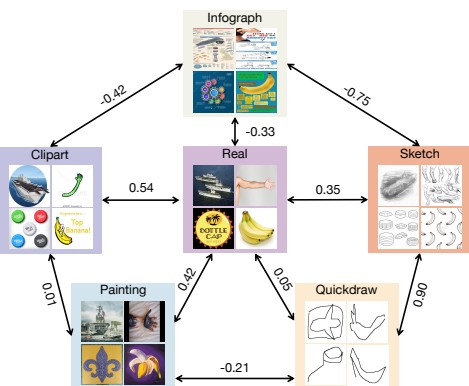

Figure 9: **Illustration of feature index similarities between different domains.** We present an analysis of cos-similarities across various domains. The results are acquired employing the GLOBAL training strategy.

Table 8: **Performance of Client2Vec on various network architectures.** We evaluate the performance of Client2Vec on the DomainNet dataset using diverse network architectures. The term 'Original' refers to the initial form of the algorithms, while Client2Vec (FEDERATED) and Client2Vec (GLOBAL) applied all three case studies. Each experiment involves 100 communication rounds, with the number of local epochs set to 5. We gauge the average test accuracy of all clients in each communication round and report the highest performance achieved across all rounds. The results are averaged over three seeds. For the VIT experiments, we use the CCT-7/3x1 models (Hassani et al., 2021).

| DomainNet | MobileNet V2 (Pre-Trained) | | | ResNet18 (Pre-Trained) | | | VIT (From Scratch) | | |
|---|---|---|---|---|---|---|---|---|---|
| | Original | Client2Vec | | Original | Client2Vec | | Original | Client2Vec | |
| | | FEDERATED | GLOBAL | | FEDERATED | GLOBAL | | FEDERATED | GLOBAL |
| FedAvg | 46.31 ±1.36 | 56.43 ±3.08 | 57.43 ±0.13 | 56.66 ±0.50 | 61.27 ±0.05 | 60.95 ±0.09 | 33.09 ±0.01 | 33.50 ±0.20 | 33.86 ±0.02 |
| FedAvgM | 45.50 ±1.21 | 58.34 ±0.01 | 57.44 ±1.04 | 57.44 ±0.42 | 61.22 ±0.11 | 60.81 ±0.18 | 33.67 ±0.56 | 34.47 ±0.20 | 34.21 ±0.11 |
| FedDyn | 45.41 ±0.89 | 51.49 ±0.17 | 53.33 ±0.26 | 58.17 ±0.61 | 61.67 ±0.42 | 59.88 ±0.42 | 29.57 ±0.40 | 31.64 ±0.13 | 31.36 ±0.12 |
| MOON | 50.56 ±0.89 | 57.03 ±0.60 | 57.50 ±0.52 | 53.80 ±0.46 | 60.76 ±0.25 | 59.90 ±0.17 | 32.29 ±0.52 | 33.58 ±0.12 | 33.73 ±0.03 |

## D.5 ABLATION STUDIES ON VARIOUS MODEL ARCHITECTURES.

In Table 8, we show how Client2Vec improves performance with different model architectures. Our results reveal that: (1) Client2Vec significantly boosts the performance of original algorithms in all settings, and (2) pre-trained models like MobileNet V2 and ResNet18 produce better results, while Client2Vec also enhances the performance of VIT models trained from scratch.

## D.6 ABLATION STUDIES ON LEVEL OF DATA HETEROGENEITY

In Table 9, we present the performance of Client2Vec in situations of extreme data heterogeneity, where each client possesses data from only two classes. The results indicate that Client2Vec significantly surpasses the original methods by a considerable margin.

## D.7 INTER-DOMAIN SIMILARITY ASSESSMENT.

Utilizing the feature index $\beta_i^f$ for clients, we quantify similarity across different domains. Figure 9 illustrates the average cosine similarities of client feature index $\beta_i^f$ between clients belonging to

| CIFAR10 | FedAVG | FedAVG + Client2Vec | FedAvgM | FedAvgM + Client2Vec |
|---|---|---|---|---|
| two classes each client | 21.35 | 66.43 | 18.05 | 63.30 |

Table 9: **Ablation studies on level of data heterogeneity.**

different domains. The results align with human intuitions, with the "Real" domain showing greater proximity to "Clipart", "Painting", and "Sketch", while exhibiting significant differences from "Infograph" and "Quickdraw". These findings validate the effectiveness of our generated client index.

