# OpenReview forum: "Client2Vec: Improving Federated Learning by Distribution Shifts Aware Client Indexing"
_ICLR.cc/2025/Conference — ICLR 2025 Conference Withdrawn Submission_

### Official Review · Reviewer_k6F8 · 2024-10-22

**Soundness:** 3
**Presentation:** 2
**Contribution:** 2
**Rating:** 3
**Confidence:** 4

**Summary:**

The paper introduces Client2Vec, a method designed to enhance federated learning (FL) by generating client-specific index vectors that capture local data distributions. This approach aims to address distribution shifts among clients, a common challenge in FL, by using these indices during three stages of the FL pipeline: client sampling, model aggregation, and local training, with the goal of improving overall training performance. Extensive experiments are conducted on datasets like CIFAR-10 and DomainNet to demonstrate the effectiveness of this approach.

However, the paper suffers from several weaknesses. The process of generating client indices and integrating them into the FL workflow introduces significant computational complexity, which could become prohibitive as the dataset size increases, especially in resource-constrained environments. Furthermore, although the paper claims that Client2Vec ensures differential privacy, the Client2Vec (GLOBAL) approach may still expose sensitive information through dataset representations transmitted to the server, raising privacy concerns.

**Strengths:**

1. Client2Vec can be combined with existing FL techniques, allowing it to enhance multiple stages of the training process without being tied to a single method, thereby offering flexibility in practical implementations.
2. The method provides a new approach to pre-training in FL, introducing client-specific representations that aim to reduce the negative impact of data heterogeneity on model performance.

**Weaknesses:**

1. The process of generating client indices and integrating them into the FL pipeline introduces additional computational complexity. As datasets scale, this overhead could become significant, potentially limiting the method’s feasibility in resource-constrained environments.
2. Although the paper claims to maintain differential privacy, the Client2Vec (GLOBAL) approach involves transmitting dataset representations to the server, which may still pose privacy risks, as these representations could retain sensitive information, making it a potential weakness in high-stakes applications.
3. The presented results are inconsistent across different datasets and training strategies (FEDERATED vs. GLOBAL). It is unclear whether the improvements depend more on the specific dataset characteristics or the chosen training strategy, which raises questions about the generalizability of the method.

**Questions:**

1. Figure 1 contains a typographical error that should be corrected.
2. As mentioned in Section 3.3, Client2Vec calculates client similarities and selects clients within the same feature domain. If the data is heavily biased, what is the computational overhead of Client2Vec compared to traditional selection algorithms?
3. Table 1 suggests that Client2Vec (GLOBAL) involves training centralized on the server using shared data embeddings. It is confusing to see improved model aggregation results in this context; a clarification of the distinction between centralized and aggregated results is needed.
4. In Table 1, there are scenarios where, within the same dataset, different algorithms show varying performance between Client2Vec (FEDERATED) and Client2Vec (GLOBAL). Sometimes Client2Vec (FEDERATED) achieves better accuracy, while in other cases, Client2Vec (GLOBAL) performs better. Is this discrepancy due to the dataset characteristics or the specific algorithms used?
5. The article could be better structured around Figures 3 and 4 to improve the flow and clarity of the presented results.
6. The analysis of experimental results should be more in-depth, focusing on understanding the reasons behind the outcomes rather than merely drawing direct conclusions.
7. For Figures 4a and 4b, two issues arise: Firstly, it is unclear which dataset is used, and this information should be explicitly stated. Secondly, there is a discrepancy in the convergence of FedDyn—while the accuracy reaches 80% after 500 rounds, it is only 70% after 100 rounds. Theoretically, the accuracy should be the same, with only the convergence speed differing. This discrepancy warrants further explanation.

---

### Official Review · Reviewer_T8HX · 2024-11-03

**Soundness:** 3
**Presentation:** 3
**Contribution:** 3
**Rating:** 6
**Confidence:** 4

**Summary:**

This paper proposes a method called “Client2Vec” aimed at addressing challenges in federated learning caused by data distribution differences among clients. Client2Vec performs data heterogeneity analysis before the actual FL training by generating a unique “client index” for each client. To obtain a representative client vector, the authors propose three loss functions to constrain the feature index decoupled from the visual encoding. Experimental results show that Client2Vec significantly improves FL performance across various datasets and model architectures.

**Strengths:**

1. The proposed method is interesting and innovative. Generating client index vectors has great potential in federated learning, especially in scenarios with domain shifts.

2. Client2Vec demonstrates significant performance improvements in client sampling, model aggregation, and local training across various datasets (e.g., Shakespeare, CIFAR10, and DomainNet) and model architectures.

**Weaknesses:**

1. An additional training process is required to obtain Client2Vec. How does this overhead compare to the downstream training tasks?

2. The resulting vectors depend on the current client dataset and the pre-trained CLIP model. The value of these vectors in real-world scenarios requires further discussion.

3. The author uses $ L_i = u^l$ . if CLIP is not well-aligned, this approach may affect performance.

**Questions:**

The method in this paper might be more accurately named DatasetVec or DomainVec.

---

### Official Review · Reviewer_gfNz · 2024-11-03

**Soundness:** 3
**Presentation:** 2
**Contribution:** 2
**Rating:** 3
**Confidence:** 4

**Summary:**

This paper introduces a mechanism before the actual training stage called Client2Vec. Inspired by Word2Vec and Domain2Vec, the authors generate a vector for each client to represent the label and feature distribution of local data which seamlessly serve as a plug-in for FL methods. Further three case studies demonstrate the efficacy of this method.

**Strengths:**

This paper conducts an information-sharing strategy with low communication cost compared with previous methods.

**Weaknesses:**

- In the abstract, it is unclear how this mechanism works.
- This paper lacks some preliminary basic knowledge, e.g., the problem definition of FL and distribution shift.
- The notions of this paper are complicated and cause ambiguity, please rearrange the mathematical symbols in the article.
- The writing of this paper should be improved.
- The extra training overhead analysis is needed for DSA-IGN.
- Is that possible to be attacked by other malicious with CLIP, because these features also get by CLIP encoder?
- What is the performance with only  improved local training, we can see from Table 1 that a relatively large performance improvement occurred with the addition of local training, which I think is probably due to the CLIP's zero-shot capability

**Questions:**

- Where is the $u^l_{i,j}$ in the Figure 1.
- Where does the DSA-IGN Workflow take place, from Figure1, it seems that all the client data is together
- Whats the detailed dimension of $D_{i,j}$, $u_{i,j}$ and $z_{i,j}$.
- Based on Definition 3.1, how to ensure that u and z are orthogonal and at the same time u and z retain as much label y information as possible? Is there a conflict?
- What are the sampling probabilities of different client data distributions?

---

### Official Review · Reviewer_bpPG · 2024-11-05

**Soundness:** 3
**Presentation:** 3
**Contribution:** 3
**Rating:** 6
**Confidence:** 3

**Summary:**

This paper proposes a novel Client2Vec method, which generates a unique client index for each client based on their data and labels before Federated Learning (FL) training begins. The client index can be used to enhance the subsequent FL training performance. The paper applies the client index to three specific case studies: client sampling, model aggregation, and local training. Extensive experiments demonstrate the effectiveness of Client2Vec in these three case studies.

**Strengths:**

1) The paper is clearly written and easy to follow
2)The paper addresses the challenge of data heterogeneity from a novel perspective and improves the training performance of FL, which is significant for FL research.

**Weaknesses:**

1) Adding an algorithm for each case study would make the presentation clearer.
2) Other classic FL baseline methods that address the challenge of client data heterogeneity, such as Ditto[1] and SCAFFOLD[2], can also be considered.
3) Although Table 3 demonstrates that Client2Vec has a shorter simulation time, the computational overhead introduced by the client index calculation needs to be quantified.
4) The authors need to evaluate the scalability of the method across various client sizes.
5) What is the specific definition of the sample index $ U_{I,j}$. The paper only provides U_{i,j}^{l} and U_{i,j}^{f}.

[1] Li, Tian, et al. "Ditto: Fair and robust federated learning through personalization." International conference on machine learning. 2021.
[2] Karimireddy, Sai Praneeth, et al. "Scaffold: Stochastic controlled averaging for federated learning." International conference on machine learning. 2020.

**Questions:**

1) As the size of client datasets continues to increase, will the additional computation introduced by the proposed method increase significantly?
2) In addition to CLIP, can other coding models be used to generate indexes? Can different coding models produce robust results?
3) Since the client index is extracted from the client data, does the index involve the client's data privacy? The author should prove that the client index does not contain the client's data privacy.
4) The global training strategy (line 256) requires uploading local data samples from clients, which contradicts the premise of privacy protection in FL. Are there alternative privacy-preserving methods to achieve global training?

---

### Note · Authors · 2024-12-04

I have read and agree with the venue's withdrawal policy on behalf of myself and my co-authors.